# Design of Audio-Augmented-Reality-Based O&M Orientation Training for Visually Impaired Children

**DOI:** 10.3390/s22239487

**Published:** 2022-12-05

**Authors:** Linchao Wei, Lingling Jin, Ruining Gong, Yaojun Yang, Xiaochen Zhang

**Affiliations:** Department of Industrial Design, Guangdong University of Technology, Guangzhou 510090, China

**Keywords:** orientation and mobility training, visual impairments, audio augmented reality, spatial cognition

## Abstract

Orientation and Mobility training (O&M) is a specific program that teaches people with vision loss to orient themselves and travel safely within certain contexts. State-of-the-art research reveals that people with vision loss expect high-quality O&M training, especially at early ages, but the conventional O&M training methods involve tedious programs and require a high participation of professional trainers. However, there is an insufficient number of excellent trainers. In this work, we first interpret and discuss the relevant research in recent years. Then, we discuss the questionnaires and interviews we conducted with visually impaired people. On the basis of field investigation and related research, we propose the design of a training solution for children to operate and maintain direction based on audio augmented reality. We discuss how, within the perceptible scene created by EasyAR’s map-aware framework, we created an AR audio source tracing training that simulates a social scene to strengthen the audiometric identification of the subjects, and then to verify the efficiency and feasibility of this scheme, we implemented the application prototype with the required hardware and software and conducted the subsequential experiments with blindfolded children. We confirm the high usability of the designed approach by analyzing the results of the pilot study. Compared with other orientation training studies, the method we propose makes the whole training process flexible and entertaining. At the same time, this training process does not involve excessive economic costs or require professional skills training, allowing users to undergo training at home or on the sports ground rather than having to go to rehabilitation sites or specified schools. Furthermore, according to the feedback from the experiments, the approach is promising in regard to gamification.

## 1. Introduction

According to the World Health Organization (WHO), there are 185 million visually impaired people across the world, of which 39 million are completely blind, including 19 million children under the age of 15 [1]. Visual impairments bring limitations, especially with respect to moving from place to place.

There are many schemes and programs designed to solve this problem or relieve the dilemma [2]. Orientation and mobility is a profession that focuses on instructing individuals who are blind or visually impaired how to safely and effectively navigate their environment. An orientation and mobility (O&M) specialist provides training that is designed to help a blind or visually impaired person develop or relearn the skills and concepts needed to travel safely and independently [3]. Orientation walking is one of the most basic needs for blind people to overcome their visual impairment, achieve smooth walking and self-reliance, and return to society. For blind people, it is important to recognize their environment using acoustic information through their auditory sense when they are walking or performing various daily activities [4]. Early orientation training aids mainly rely on professional orientation training teachers combined with tactile maps and blind sticks to guide students to complete their learning. Although these training aids are effective, only 40% of visually impaired people are trained in directional walking [5]. In addition, most walking aids and training tools on the market today are designed for visually impaired or blind adults and not for visually impaired or blind children [6]. Designing and implementing a more efficient simulated training environment and adaptive learning orientation walking aids for visually impaired children have become research hotspots [7].

Ponchillia used indoor navigation concerns and prototype feature preferences, and perceptions of the relative importance and difficulties of certain building types were identified to guide the development of accurate and user-friendly indoor navigation applications [8]. However, Ponchillia’s applications do not help people navigate the complexities of outdoor environments.

The study by Plikynas surveyed blind experts and proved that, after outdoor navigation, the second-most-important directional electronic travel assistance (ETA) feature for visually impaired people is indoor navigation and orientation (in public institutions, supermarkets, office buildings, homes, etc.). Visually impaired people need directional electronic travel aids for orientation and navigation in unfamiliar indoor environments, with embedded capabilities for detecting and identifying obstacles (not only on the ground but also at head level) [9]. Plikynas demonstrated that visually impaired people need electronic navigation devices but did not offer a specific design.

Based on these reviews, Nair et al. proposed an indoor navigation application that can guide blind and visually impaired (BVI) people through indoor environments with high accuracy, while enhancing their understanding of their surroundings. The application provides a personalized interface by considering the unique experience of BVI individuals with indoor wayfinding and offers multiple levels of multimodal feedback [10]. However, they have not adapted the training courses for visually impaired children.

Rogge et al. demonstrated improvements in gait parameters and navigation performance by evaluating 14 blind and visually impaired children and adolescents (8–18 years) and 14 visually matched individuals; no differences were observed among the blind group, the visually impaired group, and the blindfolded participant group [11]. Guerreiro et al. developed an interactive virtual navigation application (NavCog) where users can learn unfamiliar routes before visiting the environment. NavCog supports independent, unassisted real-world short-circuit navigation and has the ability to improve user performance when using accurate in situ navigation tools [12]. NavCog’s disadvantage is that it requires signals from a Bluetooth beacon, a device that is currently not widely available, making its use quite limited in parts of Asia and Africa. Meliones et al. proposed using ultrasound for outdoor navigation. The proposed solution is to use an external device that has an ultrasonic sensor and a GPS module to analyze the data received by the external sonar device in real time and decide whether it needs to emit an audible warning to the visually impaired person. The proposed algorithm can be equipped with a direction and navigation device, allowing the visually impaired person to walk safely and autonomously outdoors [13]. The disadvantage of the device is that the ultrasonic program cannot build a map and the ability to sense real-time changes in the surrounding area is poor. Unlike Meliones et al., who used smartphones, Messaoudi et al. proposed a technological solution known as smart crutch devices, which mainly used cloud computing and the Internet of Things (IoT) wireless scanners to facilitate indoor navigation. They aimed at enabling visually impaired people to move smoothly from one place to another and providing them with a tool to help them communicate with their surroundings. Recently, the application and research of adjuvant medicine have focused on computer vision. In future studies, users will learn about the emerging computer vision techniques for supporting mental functioning, algorithms for analyzing human behavior, and how smart interfaces and virtual reality tools lead to the development of advanced rehabilitation systems able to perform human action and activity recognition [14].

Training in auditory orientation is usually a part of orientation and mobility (O&M) instruction for blind people. O&M instruction is usually conducted in a real environment, and trainees should acquire auditory orientation by listening to ambient sounds experientially [15]. In China, special education in schools for the blind started at the beginning of the last century. After the People’s Republic of China was founded, this special education was incorporated into the national education system. With continuous development, special education has gradually improved, but the status of special education in China is not high. However, some of these schools are improving quickly, though still not as quickly as regular schools. In 2007, the Ministry of Education of China issued the “Experimental Plan for the Curriculum Setting of Compulsory Education in Schools for the Blind,” which offered rehabilitation courses for blind children. Comprehensive rehabilitation courses are offered in the lower grades, orientation training courses are offered in the lower and middle grades, and social adaptation courses are offered in the middle and upper grades [16].

In the United States, the Extended Core Curriculum (ECC-VI) proposed by Dr. Philip Hatlen was soon widely recognized by American schools for the blind and ordinary schools with visually impaired students. The ECC-VI includes nine significant fields: compensatory access, sensory efficiency, assistive technology, orientation and mobility (O&M), social interaction, recreation and leisure, independent living, self-determination, and career education. In terms of O&M, visually impaired students in the United States are divided into four stages: preschool, elementary school, junior high school, and high school. In the preschool years, the goal of blind children in the United States is to learn how to move toward objects, sounds, or stimuli. In elementary school, blind children practice holding a blind cane correctly or using other mobility aids to walk with simple instructions. In junior high school, blind children practice using blind sticks or other mobility aids to walk familiar routes under supervision. By the end of high school, blind teenagers need to be able to recognize and use landmarks while walking. They should be able to walk safely with blind sticks or other mobility aids and use these skills independently and appropriately [17,18].

In Europe, according to the research results of the European Blind Union, most blind people need detailed training methods, perfect laws and regulations, and complete facilities for the disabled. Commonly used devices are long canes, guide dogs, and (recently) GPS systems.

In Slovenia, the main objective of the training is to enable people to become independent and be able to travel safely in their environment. The topics covered in the training are as follows:Understand the environment: Learn about the scheme of the body, learn about the constant and moving objects in the environment, get to know the various ground surfaces, perceive noises and sounds, and perceive scents.Train the senses: Obtain a sense of movement and balance; improve hearing (sound, noise, echo); train hands, legs, and face to perceive tactile sensation; perceive aromas (recognize different scents); and identify areas depending on the destination.Grasp the walking technique: Walk with a sighted guide, walk independently without aids, and walk independently with a cane and other devices.

Slovenia has recently enacted comprehensive rehabilitation of blind and visually impaired people, regardless of the cause and age of the individual client [19].

In Denmark, children are trained by special consultants, who visit them at their kindergarten or school. Children are also offered training in general use during courses at the resource center, where they receive specific dedicated training during internal courses. As for newly blind persons, some—especially if they are in a job and needing quick and expeditious rehabilitation—are trained via lengthy courses offered by The Institute for the Blind. Almost all other training is carried out by consultants that train at the local (home) level. On-site instruction is provided at the local level as per the needs of the persons with visual impairment, but often not to the degree and length that we find is needed [20].

In Italy, the National Association of Instructors of Orientation and Mobility and Personal Autonomy (ANIOMAP), at the Institute for Research, Training and Rehabilitation (I.Ri.Fo.R), organizes special dedicated courses at the local level. Training is carried out in private facilities, at the Italian Union of the Blind and Partially Sighted (UICI) local branches, at the local health units when funded by them, at the Institutes of the Blind (resource centers), or at rehabilitation camps. The contents of a course can be summarized as follows: indoor training (accompanied walking, body protection, exploration and search techniques, use of a long white cane, principles of sensory orientation), training exercises in urban areas (exploration of residential areas, concepts of urban areas, topography, road crossings, itineraries, orientation), training routes in commercial areas (location of offices, shops, etc.; getting around in shops, offices, supermarkets, etc.), use of public transport, and training with aids (using optical and/or electronic aids for orientation and mobility; reading tactile maps, tactile compass, etc.; using optical aids, if available; and using guide dogs, if available) [21].

However, for novice trainees, training in a natural environment where actual vehicles operate can sometimes be dangerous and stressful. Furthermore, with this training method, the trainee must acquire auditory orientation because it is difficult for a novice trainee to selectively listen to essential sounds when these are mixed with many other ambient noises. To reduce the risk and stress of orientation training and shorten the training period, a new training method for reproducing an ideal sound field by acoustic simulation is presented; it is an effective method. Seki et al. developed a new auditory orientation training system for blind people using acoustic virtual reality (VR). This training system reproduces a virtual training environment for orientation and movement (O&M) instructions that produce sound sources and sound reflections and isolation, allowing trainees to learn sound localization and obstacle perception skills.

However, all of the solutions for blind people give rise to some questions: Do they have special adaptations for blind children? Do their programs help visually impaired children with walking training? Do visually impaired children still have enough interest in these traditional training methods and methods?

Meanwhile, some studies have proved that augmented reality (AR) is an up-and-coming technology, especially in disability education, and its role is becoming increasingly apparent [22,23,24]. For children with disabilities, AR technology can create an immersive learning environment, making learning more exciting and improving their learning initiative [25]. However, despite the great potential of augmented reality technology, there are few related studies on augmented reality applications for visually impaired groups [22]. Liu et al. proposed a method to develop an intuitive user interface through Microsoft HoloLens glasses: each object in the environment has a sound and communicates with the user according to commands. With minimal training, the system supports many aspects of visual cognition: obstacle avoidance, scene understanding, spatial memory formation, and recall and navigation [26]. Shrinivas et al. developed a wearable camera-based device that provides collision warnings in three directions through differential feedback from two vibrating wristbands (left, center, and right) and evaluated it in an indoor mobility course. Its mobility advantage was shown with blind and customarily sighted (NS) blindfolded individuals [27].

Suppose the immersive interaction advantage of AR can be combined with the orientation walking training course. In that case, it will improve the effect of orientation walking training for visually impaired children and expand the applications of AR.

The challenges that the project needed to face are as follows:How to design and build a safe AR environment.How to ensure that the child can avoid obstacles on the way to the destination and reach the goal safely.How to guide the travel route from a child’s perspective. How to inform visually impaired children of specific travel routes.How to use AR to enhance the user’s perception when using it [28].

This project aimed to design and develop a set of orientation walking aids based on using a smartphone, combined with Arduino intelligent hardware, as a carrier to help visually impaired children complete orientation walking training more efficiently and make the training interesting for the children. In this project, the sparse map technology of EasyAR4.0 was used as the primary technical means, combined with an immersive AR scene experience and an orientation walking space training course. Finally, an auxiliary device with the function of the orientation training course with sound scene gamification was designed to provide children with visual impairment. This research expands the applications of AR technology in special education for visually impaired children. At present, most AR application research focuses on vision-based AR applications. This study emphasizes that AR applications should focus more on auditory immersion, which is of great significance for the development of AR theory.

The reasons we chose smartphones and Arduino are as follows:(1)As a project with practical value, we expect our design to be feasible. Specifically, it means to limit the burden on users as much as possible under the premise of guaranteeing functions, especially economically, in terms of restricting the need for new equipment. Smartphones are the epitome of modern industrial production and technological progress. They integrate incredible computing power, sensing ability, and endurance on a banknote-sized board at an amazingly low cost. More importantly, computer vision solutions, such as EasyAR (in Unity), on mobile phones are at the forefront of technical societies. EasyAR (in Unity) was originally developed for multiplayer games with an indoor visual positioning functions. Therefore, the ubiquitous smartphone has become the choice for the brain of the solution we plan to design and develop.(2)To ensure that our work serves the people, we needed to consider their actual situation and the particularity of their interactions, so we planned to use external interactive devices to realize the functions beyond the intelligent core. We chose Arduino, which is friendly to both designers and developers. It has good platform compatibility, and clear and concise code, while it attaches importance to logic in the development process, and has a large number of directly adaptable sensors and interactive components, which can meet all the needs of the project at a reduced development cost.(3)The joint development of the mobile phone and the Arduino has the advantage of agility, which is particularly important for research in the prototype phase.

To evaluate the system, we conducted some experimental studies with subjects and collected process interview statistics in subsequent evaluations.

The contributions of this work are as follows:A solution for orientation walking training for visually impaired children based on AR is proposed. It uses audio augmented reality to expand the training range and improve the fun of orientation and mobility training.We designed an AR solution prototype using EasyAR, Unity, and Arduino. We created an AR audio training scenario that simulates road and social scenarios and developed design guidelines specifically for the visually impaired children.

The rest of this paper is organized as follows: Section 2 reviews the current literature. Section 3 provides an analysis of the characteristics, walking status, and needs of visually impaired children and the design of specific orientation training for children with visual impairments. Section 4 presents an experimental test of the walking aid prototype and Section 5 summarizes the work.

Figure 1 shows the research framework of this paper. The actual design process is not linear, but only the results of each stage are shown in the article, so the linear process is shown in Figure 1.

At present, there are the following solutions in terms of walking aids: tactile maps; virtual technology orientation aids; real-space, physical-simulation, 3D-sound-source aids; and AR-technology-based orientation aids. The general characteristics of these research programs are as follows:(1)Research on tactile maps

A tactile map is a topographic map that contains specific symbols and tactile cues for visually impaired students to touch. Some examples of tactile maps that can help visually impaired students are a campus’s tactile floor plan, a specific area’s (lot) tactile map, and a shopping mall’s (home) tactile floor plan. The group has a deep understanding of and familiarity with the conditions of the target environment. Ungar pointed out that tactile maps have at least two critical offerings in helping visually impaired people in spatial orientation: one is that they can help visually impaired people understand the spatial layout of an area; and the other is that they can represent the spatial environment in an environment-centric way. He also pointed out that visually impaired people obtain scene representations of the environment through tactile maps, and through orientational walking in this way, they can approach the target smoothly [29].

The current tactile maps on the market are made of microcapsule paper technology, making them easy to carry and store and highly recognizable. It is easy for visually impaired people to replace the map pattern with the internal representation in their brains [30]. There are other tactile maps on the market, such as raised-line, small-scale models made of magnet plates. These tools have been shown to improve spatial cognition in visually impaired children. However, there are limitations: the content cannot be updated once the haptic map is created. Researchers from Inria Bordeaux at the University of Toulouse have improved traditional tactile maps and designed interactive and accessible maps through augmented reality (AR) technology. Their maps combine AR projection, audio output, and tactile symbols, thus allowing visually impaired children to explore and build them. Their user research indicated that all students used the prototype smoothly and showed high levels of satisfaction. Although they are good, tactile maps for visually impaired children also involve high learning costs. If visually impaired children cannot fully understand this method, it is not easy to help them.

(2)Oriented walking aids based on virtual technology

Virtual technology orientation aids rely on users’ perceptions and control of various objects in the virtual world by using sensing devices, such as helmet-mounted monitors, data clothing, data gloves, and force feedback, to create an immersive experience. Because they have no visual experience, visually impaired people can perceive information only through non-visual channels. Virtual reality technology allows visually impaired groups to perceive spatial information in the virtual environment mainly through hearing and touch. Virtual reality technology is immersive, interactive, and safe. In such a system, blind people can explore and navigate freely and happily. They can form their mental maps of the environment efficiently using virtual reality. However, VR equipment is relatively expensive and difficult to promote. Virtual technology is not suitable for all places, and if misused, it can bring about side effects. For example, in a secret room in a virtual space too much sensory perception may cause the subject to feel uncomfortable, causing the subject to distrust the system. In addition, in reality, blind people place great importance on using echo cues. In contrast, audio cues in virtual environments have not been addressed, causing distortions in virtual environments in this regard.

(3)Real-space, physical-simulation, 3D-audio-source auxiliary equipment

This equipment mainly simulates the sound in the natural environment through hardware sound sources in different directions in real space to train the orientation abilities of visually impaired children. The traditional real-space, physical-simulation, 3D-sound-source auxiliary equipment is mainly based on the 3D sound source space of the speaker array, and it is not easy to flexibly control the position of the sound source—the simulated sound source [31,32,33,34,35,36,37,38]. However, it is difficult for most visually impaired groups to use similar conditions for orientation training, making it challenging to popularize this piece of equipment among the public.

(4)Oriented walking aids based on AR technology

Augmented reality (AR) technology can combine virtual objects (virtual images, models, sounds, etc.) in natural scenes and allows users to interact with virtual objects in real space. This interactive method based on the natural world and enhanced by virtual data provides students with a learning space that can be explored in a way that is closest to natural interaction. AR is a cutting-edge technology, but some of its characteristics coincide with some points of education theory. 

At present, many studies have found [24,25] that these characteristics of AR technology endow it with unlimited development potential in the field of education. However, there are few studies and applications for disabled people, especially for visually impaired children.

Researchers such as Lei Zhang of New York University have improved and developed a mobile phone application that combines AR and VR. Due to mixed reality (the fusion of VR and AR), the project is also called Mixed Reality Blind Combat (MR cane) [26]. The program simulates a blind cane in virtual reality by using a mobile phone with a selfie stick, applies AR technology to track real-time poses in the real world, and then synchronizes with the blind cane in the virtual environment. The moment the virtual blind cane comes in contact with an object in VR, the user gets to know it through auditory and tactile feedback [26]. According to the author’s experimental results, it can effectively help visually impaired groups to understand interactions with virtual objects and explore the 3D virtual environment. Although Mixed Reality Blind Combat (MR cane) is effective, it still has shortcomings because the audio is simulated 3D-space audio (made via algorithms) and the location of the sound source is not accurate enough and needs to be improved.

Visual impairment refers to the congenital or acquired impairment of visual function to a certain extent. Visually impaired children have lost direct access to visual information and can perceive external information only through the remaining senses. Hatwell found that visually impaired children face bigger challenges than ordinary children in spatial perception, especially in distance-related depth perception and accurate perception [25]. Unlike children with normal vision, visually impaired children mainly rely on their tactile and auditory senses to perceive external information with lower richness. The lack of visual experience makes it difficult for visually impaired children to form visual representations. As a result, visually impaired children in lower grades show the characteristics of mainly mechanical memory, leading to weak working memory. However, visually impaired children have improved auditory awareness due to many factors, such as intensive exercise and the reorganization of neural attention mechanism in the brain. Moreover, it is difficult for them to use visual cues because either they congenitally lack visual experience, or their memory of the visual experience has declined. Therefore, visually impaired children face remarkable difficulties in the situational awareness [25].

To allow visually impaired children to perform activities, more reliable and safer barrier-free facilities are needed that encourage them to explore the environment. Accommodation should be provided for children to explore through non-visual senses, through which they can learn to move, practice their recognition, and memorize the environment [39]. 

## 2. Materials and Methods

Our work aimed to examine the possibility of reconstructing audio-AR-based O&M orientation training. To design and provide a hybrid solution, we started by identifying the main problem experienced by blind subjects, as shown in Table 1. To better empathize with the situational awareness of visually impaired children, simulated training was adopted to further explore the skills and training styles they want most.

In the study, eight visually impaired children and three adult guardians of visually impaired children were invited. Table 2 presents summaries of the interviews.

Due to the difficulty of contacting visually impaired children offline during the epidemic, the authors strengthened their awareness of orientation training for visually impaired people by participating in the blind volunteering activities of the Guangzhou Association for the Blind and observing the work activities of five adult visually impaired persons.

By observing their behaviors and interviewing them, we realized that with white canes, visually impaired people are capable of moving as fast as normal-sighted people in familiar environments. However, when it comes to unfamiliar sites, visually impaired subjects are prone to acquiring external assistance.

To experience the challenges in orientation, we also used simulated training. Keeping that in mind, we summarize the interviews and observations in three dimensions: basic, travel, and willingness to use and outline the relevant findings of the simulated content. The summary is as follows:Most of the users interviewed in the study were blind, and a few had low vision. Most of the interviewees were congenitally blind, and a few were blind due to acquired diseases. Nearly half of them had never received orientation training. It can be seen in the interview records that the spatial cognition of visually impaired children who have received orientation training is more effective than that of visually impaired children who have not received such training. Because the visually impaired children were not guided in their early orientation walking learning, some visually impaired children are unable to travel independently, which has a severe negative impact on their spatial cognitive ability and will have a severe negative impact on their future growth. Parents of most visually impaired children do not allow them to go out alone, which may stifle the essential living ability of visually impaired children and further widen the gap between visually impaired children and other children.Most of the respondents had not received orientation training before the age of 7 years. Even after training, the trainees will still have problems, such as inaccurate orientation judgment and the inability to walk straight in the early stages of training. Due to safety factors, visually impaired children can only train in a school environment. When they enter the later training period, problems such as boredom and low learning efficiency appear because they are too familiar with the campus space and have no extra space for further exploration. Visually impaired children generally like to learn orientational walking with specifically designed games. Most visually impaired children are reluctant to practice on the road by themselves because of their fear of the unknown external environment.The interviewed subjects are highly receptive to new orientation aids. However, they are still worried about the immaturity of the new technology and the lack of apparent training effectiveness. For assistive devices for orientation walking, most subjects believe it is necessary to simulate different kinds of roads; in addition, the device should make the training process more enjoyable. Most visually impaired children hope that friends or parents can join their training to improve training efficiency and improve their family and friends’ understanding of them.Spatial positioning and angle positioning are critical for visually impaired children to orient themselves in walking, thereby determining whether the visually impaired can reach their destinations smoothly.Parents of visually impaired children are expected to devote more energy and financial resources to educating and raising them. Most of the low- and middle-income families urgently need sufficient support from the government and society [16]. The families are price sensitive, and the cost-effectiveness, or feasibility, of the solution should be fully considered in the design.

## 3. Design of the Orientation Training

According to the analysis of the characteristics of the visually impaired children and the needs of orientation training described in the preceding section, we designed and developed a set of orientation training systems using a smartphone as a carrier with separate audio devices to provide visually impaired children with more exciting and efficient orientation training. We combined immersive AR and key features from a conventional orientation training course, took EasyAR4.0 for localization, and adopted the audio prototype hardware for the voice to design the situational gamification training course, which is expected to deliver an entertaining experience in training. Figure 2 presents the designed concept.

The design scheme also considers the social difficulties visually impaired children face, providing them with a way to invite friends or relatives to participate in their own training, as shown in Figure 3. Visually impaired children may experience a moving sound source by wearing amulet sound source accessories, as can their friends or relatives who join the training. For example, if visually impaired children choose to cross the road, they can simulate walking around passing vehicles. Then, the AR orientation training application tracks the amulet sound source accessories, which can describe the relevant road conditions and give relevant task instructions so that the visually impaired children can better complete the orientation training. This not only improves training efficiency, but also enhances the attraction of event participation.

The augmented reality module for orientation walking for visually impaired children is mainly based on the visual SLAM of EasyAR. It adopts the vision-based positioning scheme and plans the path by marking the sound source point of interest (POI), as shown in Figure 4. It is divided into two parts: online and offline. In the online stage, the area to be trained is first scanned and stored through the mobile phone and then the observed data are uploaded to the cloud to build a 3D visual map. When the phone revisits the same space, it can obtain map information about the training area by simply accessing the pre-built visual map. In the offline stage, we need to use the POI, which represents the sound source point that needs to be reached in training. Then, we hang or install corresponding appropriate hardware at different sound source positions and allow the mobile phone to track the hardware sound source in real time to simulate the complex social environment of the training area in real time. Finally, we improve the spatial cognition of visually impaired users during the game by guiding them to find additional sound hardware. At the same time, the guide allows visually impaired users to experience the course in an immersive AR sound-effect environment and a completely different orientation finding game tasks. The design also has the advantages of a simple hardware structure, convenient operation, and rapid implementation.

The reasons why we chose the Arduino and smartphones are as follows:Facilitate the follow-up code development and research. EasyAR is built on the Unity engine, so it needs a platform with high computing power portability to support Unity. We chose the Arduino to program the size of the sound and change the sensor. It was relatively easy for our team to understand Arduino and smartphones. However, it is more challenging to develop if we use STM32 or 51 microcontroller and other microcontroller or FPGA development.The external expansion of the development environment of Arduino and mobile phones makes it easy to find highly compatible components, thus reducing the difficulty and cost of development.Almost every Chinese has a smartphone. Compared with development boards such as FPGA circuits and NVIDIA Jetson Nano, the cost of a smartphone APP and an Arduino development board is extremely low, and a smartphone is easy to use and promote.

## 4. Experimental Plan and Process

### 4.1. Experimental Procedures, Results, and Interviews

We used comparative experiments to verify the effect of the training. The first experiment in this study involved an 8 × 4 m^2^ experimental scene in an indoor environment. In the experimental site, more than 20 obstacles of unequal volumes were placed. These included tables, chairs, and daily household necessities to simulate reality and enrich the experimental system. Figure 5 shows the specific experimental scene.

The tests involved the following steps: (1)Install the designed and developed AR orientation training application on the smartphone.(2)Use the mobile phone to check the outer surfaces of the objects in the experimental site to build up a sparse map.(3)Mark the POI of the sound source to determine the position of the target sound source and the position of the interfering sound source and situate the corresponding sound source hardware in the experimental scene.(4)Designate the movement path.(5)Generate the correct walking path, as shown in Figure 6.

Specifically, the phone we used in the experiments was a Huawei P20, the SDK for development was Unity3D, and the onboard map builder we chose was the EasyAR platform.

The main experimental steps were as follows: (1)Normal-sighted children were invited to first familiarize themselves with the relevant commands of the AR orientation training application.(2)The children were asked to wear a blindfold 10 min prior to the training tasks with the app. Each child was asked to participate in three trials. Here, we regarded the training as successful if the child could navigate to the destination in at least two trials.(3)The experimental and interview results were collected and analyzed. The interview questions are listed in Table 3.

In this experiment on the effectiveness of the proposed training, all six blindfolded participants were able to reach the target sound source. However, they did show a certain degree of drift in the experimental results. According to our interviews with the six subjects after the event, all six members knew they had deviated away from the sound source. There we two issues: (1) being more sensitive to car sound sources, and (2) choosing a relatively sizeable-in-relative-error route to avoid collision with the sound source. This phenomenon indicates that each user empirically predicted the relative position of the sound source in space and could react to it. Therefore, as long as the experimenter can avoid obstacles and reach the target sound source, the experiment can be regarded as a success.

The feedback scores were collected by interviews, as shown in Table 3. After sorting the average score for each question, we prepared the results in the form of Figure 7. After sorting the scores of the interview records and statistical tables, we found that the participants tested generally believed that the experiment was interesting and immersive and that it could guide them to the target sound source, but the stability of the application needed to be improved. One of the subjects believed that the software was good in terms of its effectiveness and perception and it also performed well in the appearance aspect. However, one participant thought that there were too many kinds of sounds in the experimental scene and it was difficult to identify each.

### 4.2. Inter-Group Comparison Experimental Process and Results

After our first experiment, to further verify the effectiveness of the orientation training, we conducted an inter-group comparison experiment. Two groups performed the comparative experiment. One group of children used the prototype intervention training of the orientation training aid, and the other group did not receive the aid. The test site of the verification experiment was an outdoor area of 8 m × 3 m. We built a simulated road scene in this area and prepared a tram and a cart. To better simulate the actual road conditions, the author placed several stools on the side of the road to act as vehicles and obstacles on the road. The specific arrangement is shown in Figure 8 and Figure 9.

Before starting the experiment, we needed to prepare for the investigation as follows:(1)The researcher first used the AR orientation training application prototype to scan objects on the experimental site to construct a 3D visual map.(2)After scanning the map, we marked the POIs, determined the starting point, and generated navigation paths. Figure 5 and Figure 6 display the navigation path.(3)We turned off the navigation sound because the software only needs to be used to record the subjects’ movement path.(4)We informed the staff of issues and safety precautions. The participants were blindfolded for 10 min.

(1)Two staff members who participated in the experiment drove electric cars and bicycles back and forth in the simulated space. One of the staff members parked the tram on the road and honked the horn.(2)The tested children came to the simulated road and completed the task of crossing the road.(3)After the experiment, we interviewed the subjects about the usability scale.(4)We analyzed the experimental results.

During the experiment, if the children could walk to the target position and avoid all vehicles and people in the simulated scene, the experiment was considered a success. However, if a subject encountered any vehicle or failed to reach the target position during the experiment, the experiment was considered a failure.

A total of 12 children were involved in the inter-group comparison experiment. Six of the children were subjects who had undergone the training intervention in the previous investigation, and the other six were untrained. Ten minutes before the experiment, the children wore a blindfold to mimic the vision of a newly blind child.

In the experiment, we controlled a simulated traffic light and produced different audio frequencies. When a traffic light emitted a high-frequency audio signal, indicating a green light, the children could cross the road. If the children were crossing the road with no pedestrians or cars in the middle, they were more likely to succeed. If there was a red light, the simulation environment had cars simulating the actual scene, and the participants were more likely to fail. Figure 10 shows two photographs of the comparison experiments between groups.

In this experiment, participants in the intervention group reached the finish line during the green light. The authors screened the refined relevant experimental data and sorted the suitable and effective mobile data into the path graph shown in Figure 11. In the figure, purple represents the planned path, the red box represents the end range, and other colors represent the movement paths of six subjects in the intervention group. All blindfolded participants could move from the starting point to the endpoint, and some encountered roadside obstacles. The subjects were informed that they had suddenly lost their direction in the middle of the test and had finally found their original direction after coming in contact with obstacles.

Compared with the trained children, the children without intervention training were less able to complete the task. Although four of the six subjects reached the final destination, the subjects repeatedly bumped into obstacles during walking. Three children ran into pedestrians and vehicles crossing the street during a red light. The fourth subject reached the destination by bypassing the barrier, but the subject seriously deviated from the actual walking path, even beyond the experimental range. The fifth and sixth children walked back and forth near the starting point but did not reach the end area. Figure 12 shows the paths of the six children. Among them, purple represents the planned path, the red box represents the safe area, and other colors represent the paths of the six subjects without intervention training. After the interview, the subjects learned that they could not judge the specific location, did not know the sound difference of traffic lights, and had weak spatial perception ability.

On analyzing the experience questionnaire results of the two groups of children (Figure 13), we found that the average score of the experience scale for children with intervention training was higher than that of the children without intervention training.

## 5. Conclusions and Discussion

In this work, we first discussed the relevant research in recent years and pointed out their shortcomings in the corresponding fields. Then, we developed an orientation training framework and application for a smartphone to help visually impaired children in orientation training in O&M. Next, we conducted questionnaires and interviews with visually impaired people, especially impaired children. Next, we used the advantage of immersive AR audio with training courses and designed an auxiliary device with AR-audio-scene gamification orientation training to provide visually impaired children with more interesting walking assistance training. It innovatively combines AR audio and physics to successfully improve the efficiency and entertainment of O&M training. After designing the equipment and curriculum, we recruited children with normal vision, conducted experiments after blindfolding them, and proved the effectiveness of the system through the experiments and post-test interviews.

To conclude, in this work, we propose an orientation training method based on mobile computed audio augmented reality. This approach is based on the ubiquitous smartphone and on Arduino, which has superior usability and feasibility, low equipment and scheme costs, low requirements for professionally skilled coaches, and flexible demand in terms of the place of training.

Compared with other orientation training studies [4,5], our method makes the whole training process flexible and entertaining. At the same time, this training process does not involve excessive economic costs or require the skills of professional trainers, allowing users to undergo training at home or on the sports grounds rather than going to rehabilitation sites or specified schools, compared with the well-known Seki’s audio augmented reality orientation training solution [22].

The equipment and curriculum we designed have some shortcomings. First, although the simulated environment constructed by the training equipment for visually impaired children provides a deeper understanding of the navigation and development of visually impaired people, a virtual training environment for visually impaired children cannot fully imitate the actual road training. Still, most trained experimenters believe that the project and the equipment achieved excellent results. Secondly, the system itself requires further testing and optimization. For example, if the light changes significantly and the moving range of the object is large, it may cause a slight position shift. In addition, due to time constraints, we could simulate only some functions of the designed solution for in-time testing, and the experimental conclusions do not fully reflect the experiences of all users.

In recent decades, virtual reality and augmented reality have been widely used as promising technology, providing relatively feasible and inexpensive solutions for enhancing activities such as nursing care and physical rehabilitation of disabled people. However, visually impaired people have little access to the benefits of this technology. In addition, people facing inconveniences in social life also need the attention of relevant policymakers, rehabilitation experts, researchers, and developers. The augmented-reality-based orientation training designed in this project can more effectively train visually impaired children in terms of orientational walking ability. Visually impaired children can invite parents or friends to wear audio accessories to participate in the training to increase the fun of training and enhance friendship. Orientation walkers use a smartphone as the primary carrier. The cost of the whole set of equipment is low even after the device is combined with audio accessories, which is in line with the affordability requirement of ordinary families with visually impaired members in China and is suitable for large-scale promotion. The AR orientation walker conforms to the development trend of AR technology, expands the application scope of AR technology in visually impaired people, and contributes to the development of AR theory.

Due to time constraints, the various sound source accessories we prepared were not enough to simulate complex road scenes. At the same time, considering factors such as the COVID-19 epidemic situation and time constraints, the main subjects of this research were children and college students with normal vision and only a limited number of visually impaired children were tested and interviewed. Although blind children have better hearing and touch abilities than normal blindfolded adolescents, we still need and will continue to conduct future experiments on blind children to demonstrate the effectiveness of our devices.

Another interesting result that was not an intended goal of the experiment is as follows: when the participants were interviewed, many of them said that they felt the experiment and training we conducted were like a fun game. Although this was not part of our interview question, it made us feel like games are a possible way to train such children. Although gamified tasks are not the core of the study, we found in the course of the study that entertaining or gamification content can easily stimulate children’s curiosity and participation enthusiasm.

## Figures and Tables

**Figure 1 sensors-22-09487-f001:**
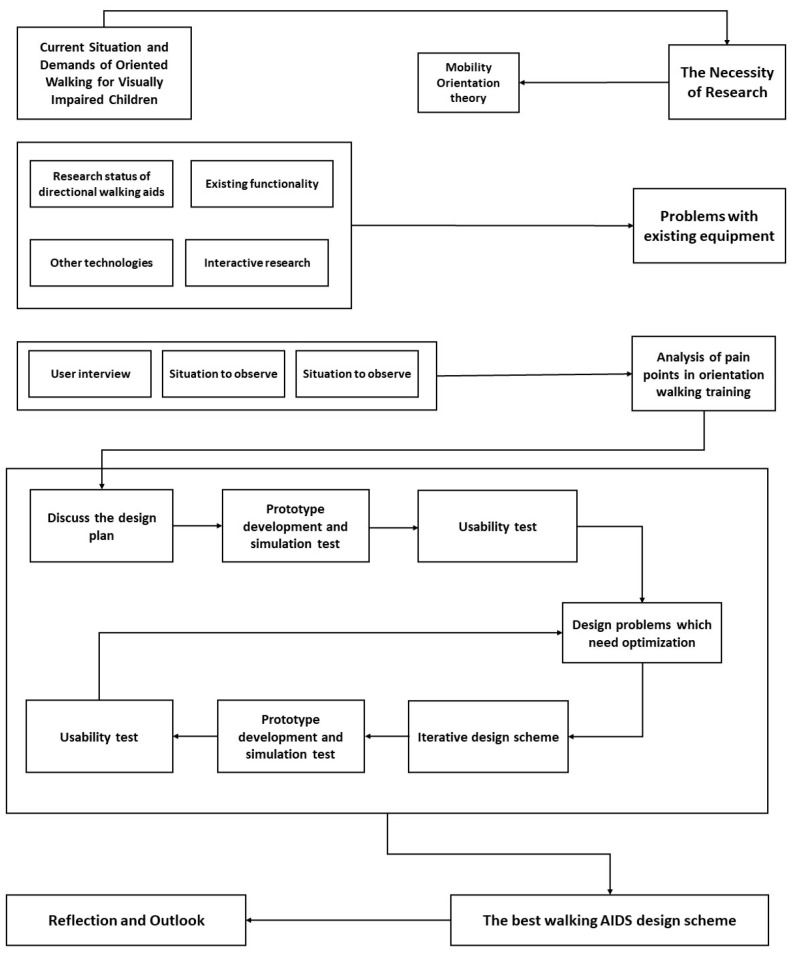
Research framework.

**Figure 2 sensors-22-09487-f002:**
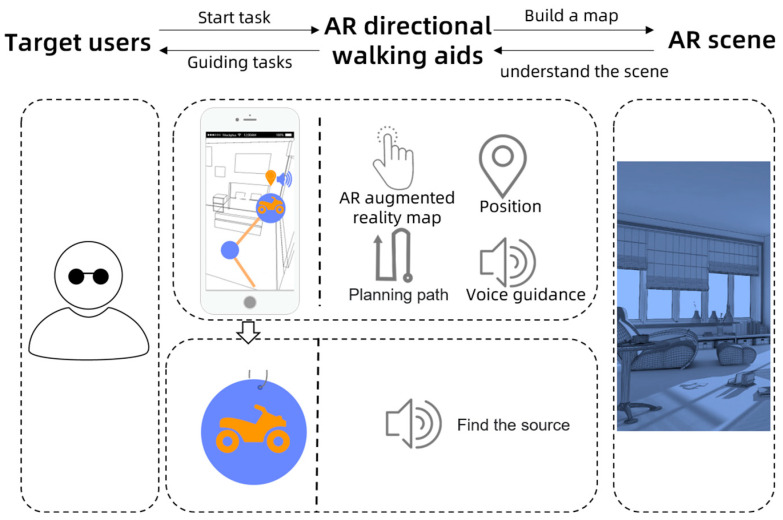
The design concept of the orientation walking aids.

**Figure 3 sensors-22-09487-f003:**
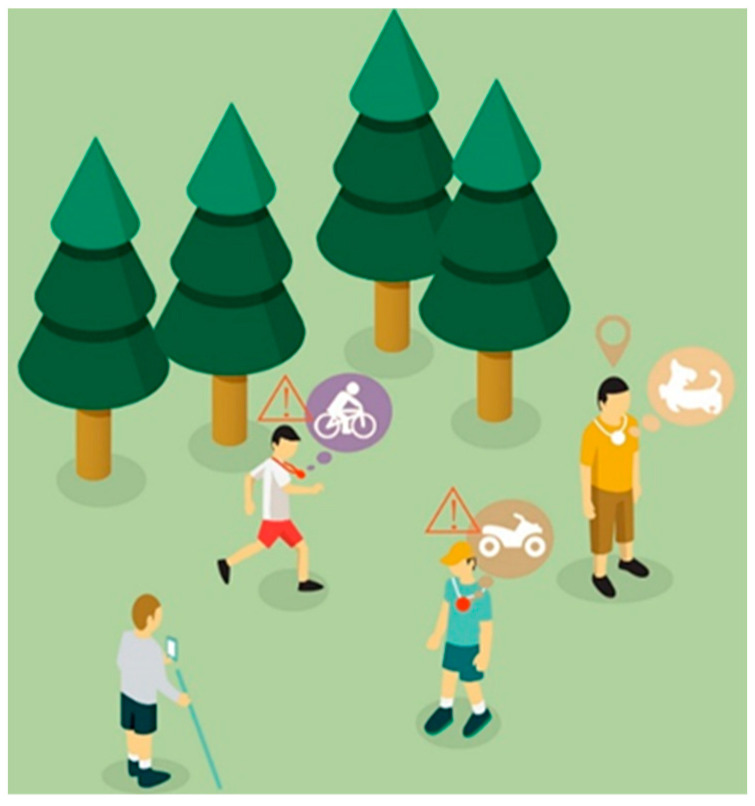
Scene diagram of the use of mobile audio accessories.

**Figure 4 sensors-22-09487-f004:**
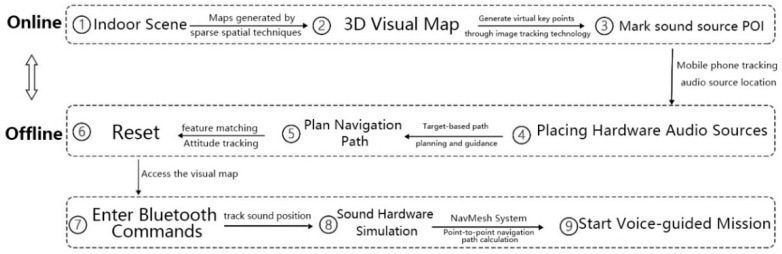
Design concept drawing of orientation aids.

**Figure 5 sensors-22-09487-f005:**
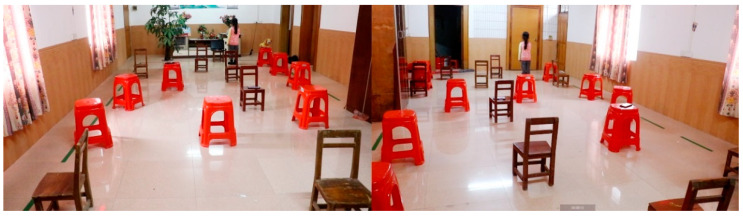
AR orientation aid equipment study site.

**Figure 6 sensors-22-09487-f006:**
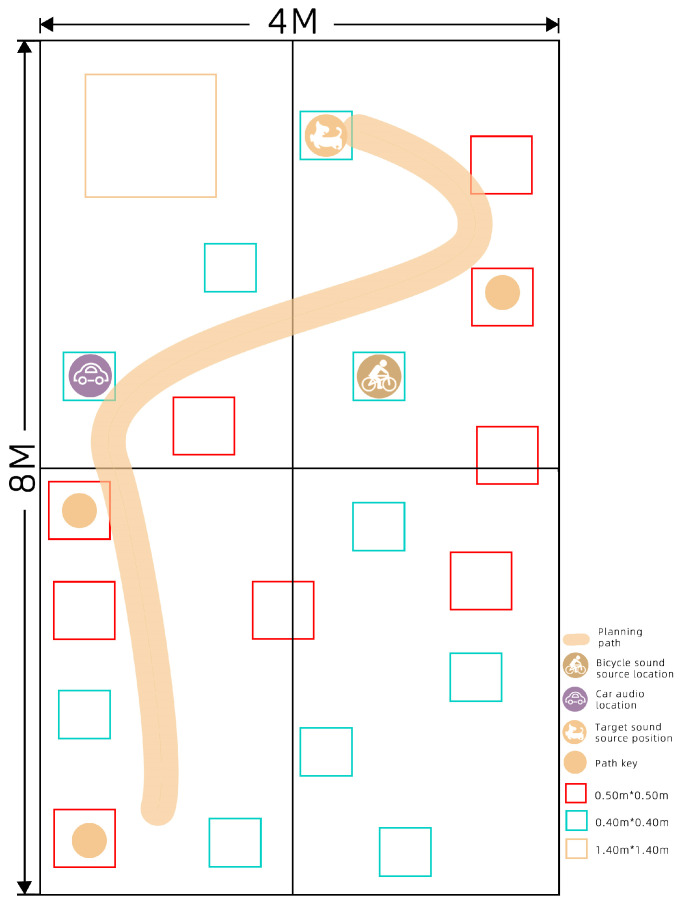
Schematic diagram of the experimental site.

**Figure 7 sensors-22-09487-f007:**
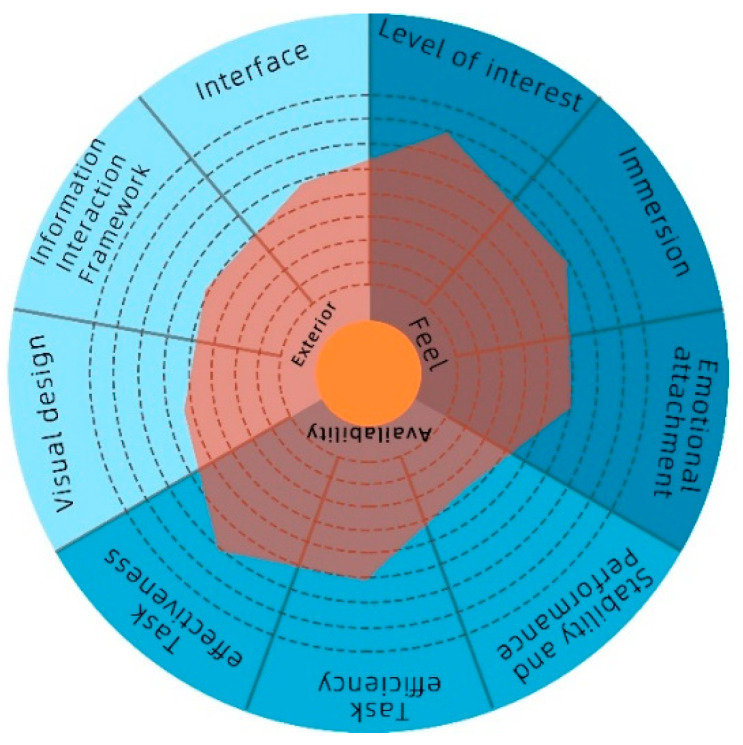
Radar scoring of the posttest evaluation.

**Figure 8 sensors-22-09487-f008:**
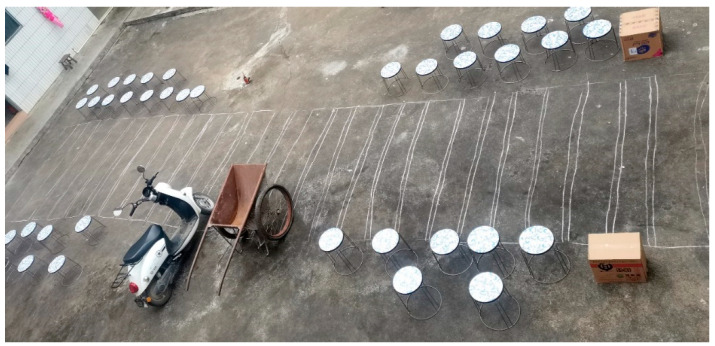
Scene diagram of the inter-group comparison experiment.

**Figure 9 sensors-22-09487-f009:**
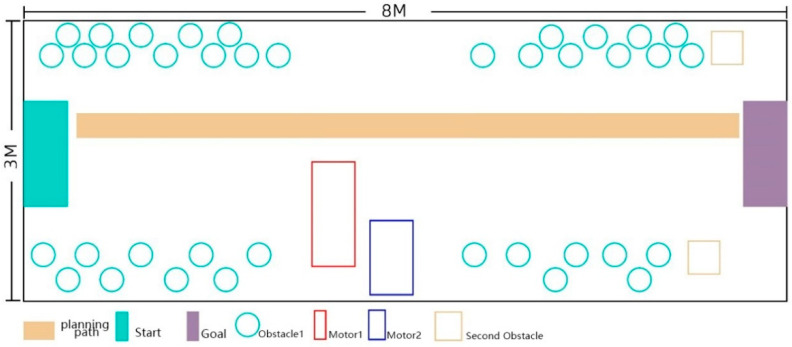
Path map of the inter-group comparison experiment.

**Figure 10 sensors-22-09487-f010:**
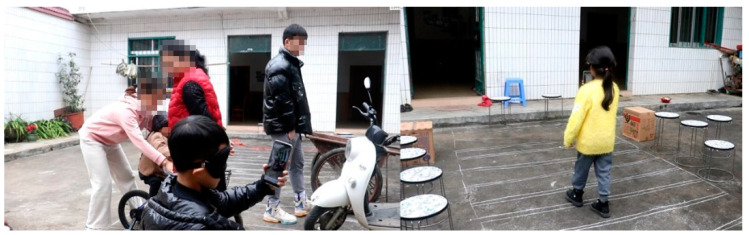
Photos of the inter-group comparative experiment.

**Figure 11 sensors-22-09487-f011:**
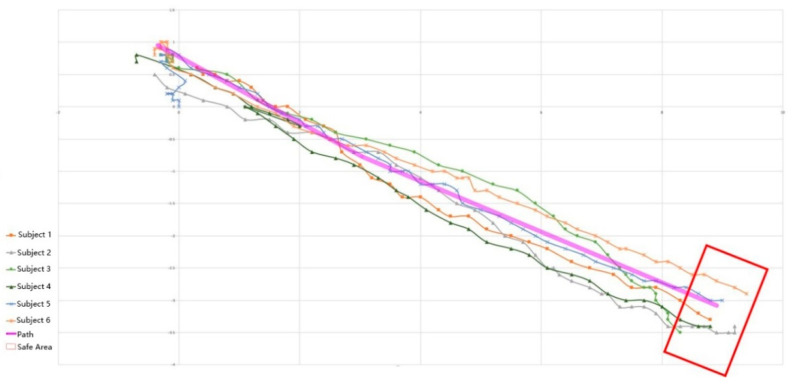
The path records of the intervention group crossing the road.

**Figure 12 sensors-22-09487-f012:**
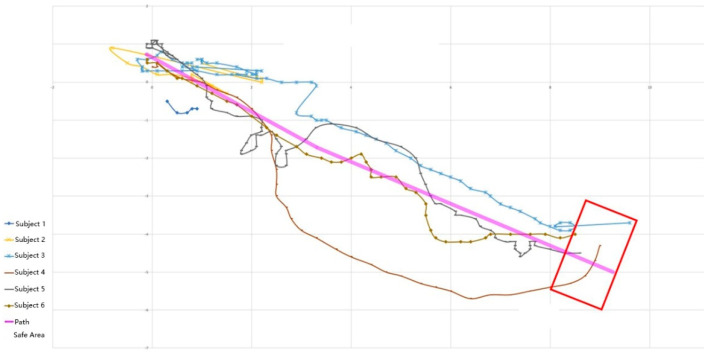
The path recordings of the non-intervention group crossing the road.

**Figure 13 sensors-22-09487-f013:**
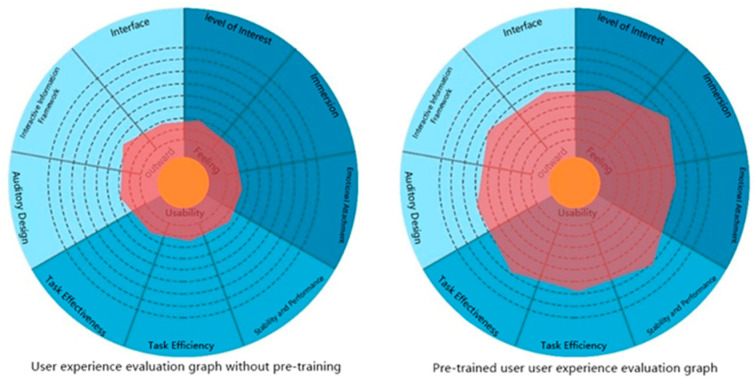
Quantitative user experience evaluation maps of crossing the road for the non-intervention group and the intervention group.

**Table 1 sensors-22-09487-t001:** The dimensions and questions of user interviews.

Dimension	Question
Basic condition	When did you lose your eyesight? What is the extent of the damage? Have you received formal systematic orientation training? Do you know how to use a smartphone? What kind of smartphone will you use? Do your parents allow you to go out alone? Do your parents know how to instruct visually impaired children in O&M training?
O&M Orientation Training condition	6. When was your first experience of systematical orientation training? 7. What kind of environment do you choose for your daily orientation training? 8. What situations are you most worried about during orientation training? 9. Are external aids applied in the orientation training? What are they? 10. What difficulties have you encountered in the process of orientation training, and how do you solve them? 11. Which form of orientation training are you most interested in? 12. Do you train alone during orientation training? 13. Are you willing to go to the social environment for orientation training?
Willingness to use alternatives	14. If there is an assistive device that can provide an immersive orientation training and can gamify the training course, would you be willing to use it? And why? 15. What features would you like this orientation training to have? 16. Would you be willing to allow your family or friends to join your orientation training process? Why?

**Table 2 sensors-22-09487-t002:** Summaries of user interviews.

Dimension	Question	Content Mention Frequency (/Person-Time)	Content Mention Frequency (/Person-Time)
Basic condition	1	Vision impairment occurrence before age 5	8
		Vision impairment occurrence after age 5	3
		Total blindness (blind people)	9
		Visually impaired (people who are visually impaired except for blindness)	2
Basic condition	2	Received orientation training	6
		Not received any orientation training	5
Basic condition	3	Never used a mobile phone	1
		Use an Apple smartphone	2
		Use an Android smartphone	3
Basic condition	4	Not allowed, too dangerous	7
		Allowed, very relieved	1
		Allowed, but will be watched from a distance	3
Basic condition	5	No idea	6
		A little bit, but no idea on its practice	3
		Skilled, practically experienced	2
Orientation Training condition	6	Never	4
		Contact before 7 years of age (including 7 years of age)	2
		Contact after 7 years of age (excluding 7 years of age)	5
		Visually impaired (people with impaired vision other than blindness)	2
Orientation Training condition	7	Home	2
		School	5
Orientation Training condition	8	Worry about finding the way	1
		Worry about training boredom	3
		Worry about being laughed at by discerning children of the same age	3
Orientation Training condition	9	Only white cane	1
		Only tactile map	1
		Both white cane and tactile map	5
Orientation Training condition	10 Multiple choice	Inaccurate orientation judgment	5
		Inefficient training	4
		Do not know the effect of training alone	3
Orientation Training condition	11 Multiple choice	Game	7
		Professional’s coaching	4
		Practice alone	2
Orientation Training condition	12	Training alone	2
		Multiplayer training	5
Orientation Training condition	13	Willing	2
		Not willing	9
Willingness to use	14	Willing	7
		Not sure; depends on usage	3
		Prefer to use a blind stick	1
Willingness to use	15 Multiple choice	Audio gamified function	8
		Training sense of orientation and spatial perception	9
		Training not only hearing but also smell and touch	7
		Training replay or review	7
		Simulated socialized activities	10
Willingness to use	16	Yes, it is likely to help subjects to get used to family and friends	8
		No, do not want to bother family and friends	1
		Not sure; it depends	2

**Table 3 sensors-22-09487-t003:** Quantitative user experience questionnaire in post-experiment interviews.

Question	Fractional Value						Average Score
	A	B	C	D	E	F	
How would you rate the efficiency?	7	7	4	7	6	6	6.17
How much do you think this prototype can quickly guide you in the task?	6	7	5	6	5	5	5.67
How would you rate the timely response?	5	5	3	4	6	5	4.67
How would you rate the entertaining experience?	7	7	5	7	6	6	6.33
How would you rate the immersive experience?	7	6	5	7	6	6	6.17
How would you rate the degree of satisfaction?	7	6	5	7	5	5	5.83
How would you rate the accuracy of the instructions?	6	6	3	7	5	7	5.66
How would you rate the degree of understandableness?	6	7	5	6	4	4	5.33
How would you rate the degree to which the interface layout meets your expectations?	6	6	4	7	5	5	5.50

## Data Availability

The data presented in this study are available on request from the first author.

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
