# Peer review of "Design of Audio-Augmented-Reality-Based O&M Orientation Training for Visually Impaired Children"

_sensors, 2022, doi:10.3390/s22239487_

Round 1

Reviewer 1 Report

It seems to me an interesting line of research that contributes to this area.

Author Response

It seems to me an interesting line of research that contributes to this area.

Response: Thank you very much for your time in reviewing the manuscript and your encouraging comments on the merits.

Reviewer 2 Report

The authors designed and developed an orientation and mobility training aid for visually impaired children. The importance of the topic is well defined at the beginning of the paper, and then the reader is systematically introduced to the research process in a well-structured manner. The development and research presented aims to make the practice of orienteering for visually impaired children more effective and interesting. In the presentation of the project, it is given that an Arduino-based smart smartphone, and specifically a mapping technology called EasyAR4.0 was used to develop the project for the research. At this point, a question has already arisen in my mind regarding the choice of devices, because I miss the reasoning behind the choice of devices in the study. I would like to see a more detailed description of the IT devices support.

Various projects and studies have been reviewed with the aim of confirming the relevance of the experiment and development. To this, the researchers undertook a questionnaire survey and interviews with visually impaired people. In my opinion, they provided a very good foundation by using a variety of research methodological tools to take evidence. However, what struck me was that Asian and American realised cases were mentioned in the study. I find it interesting that no European situation or case could or would have been mentioned? Why?

O&M is a complex behavior, which much taps into the domain of decision making. How do the authors relate their research to that of made regarding decision making? E.g. see http://acta.uni-obuda.hu/Pejic_StanicMolcer_108.pdf

After the examples were given, it was reported that they had developed a prototype of an innovative device using AR voice source tracking technology, thus providing evidence of the uniqueness of the device. Following development, they intended to conduct an experiment with visually impaired children. A further question that influenced the experiment arises here. The researchers carried out the experiment with blindfolded children, i.e. not visually impaired, but with normal vision. Wouldn't it have been more effective and realistic to expect more realistic results if they had actually used visually impaired children to conduct the experiment with the new device? After all, normal children have different perceptual abilities and different levels of development than visually impaired children, because those with congenital problems often have different abilities that are often more developed than their normal partners.  In my opinion, further research would be worthwhile to actually carry out the training developed with visually impaired children in order to achieve truly realistic results.

Ar can be technologically challenging for the users. How did authors assess the quality of the AR?

The study presents an orientation, training framework and application for a smartphone to assist visually impaired children in O&M orientation training. In the description of the developed aid, gamification is mentioned several times, which the authors use to draw attention to the development of fun, experiential training. Despite the multiple mentions, throughout the reading, I was waiting to see where the characteristic of gamification was more fully explained, where it was operationalised? In conducting this experiment, it would have been good to record experiences that point to experiential education. They wrote that visually impaired children usually like to learn to walk with games specially designed for this purpose. It would be good to complement this idea with concrete examples of playful exercises. Without such experiences, knowing AR technology, it is left to the reader to determine the quality and form in which gamification is manifested in the experiment or in real-world use with this potentially forward looking solution.

Finally, summarising the questions I have identified with the aim of adding to and answering these questions within the article to make it more complete:

Why were Arduino-based smartphones used for the research? What are the features of Arduino that made them choose it? It might be worthwhile to make a small comparison with other phones to justify the choice.

Do you know of similar solutions used in Europe or tested outside America and Asia?

I would have found it worthwhile to conduct the experiment not with blindfolded children, but with children with real visual impairments. At the very least, it would have been good to formulate the reasoning behind this.

For the often mentioned gamification, could you list applied game-tasks as an example? Which ones were used during the training? Or what did they mean by the effectiveness of gamification during the research?

Author Response

Dear Reviewers,
    We have responded to and corrected your question.Please see the attachment.

Reviewer 3 Report

The manuscript needs one more round of editing in terms of English and scientific writing style and some parts of the paper are misleading or have no logical order or flow. The paper must go through a rigorous editing and proofreading process before resubmission.

The 2nd limitation of the manuscript is that it lacks critical related work. The historical perspective should be discussed. The proposed study is not critically evaluated and compared to the related work/state-of-the-art and is not identified and discussed its drawbacks and limitations. Thus, it is not easy to assess the real contribution of the paper in the field and how much is efficient in the proposed study compared to related works. A clear assessment of the contribution of the authors when compared to existing approaches should be given. Thus, It also becomes necessary to propose future actions for the improvement considering the studies carried out. It is advised to revise the Abstract and Conclusion. Both sections should be consistent in terms of Proposal, Problem statement, Results, and future work. As in the current format, both are opposing each other a little.

Author Response

(The authors gave the same response as above.)

Round 2

Reviewer 2 Report

The authors provided excellent evidence of the seriousness and dedication of their work. Their answers to the questions are largely satisfactory. Necessary and sufficient additions have been made in their manuscripts in a well-written and well-edited manner. In my opinion, the result is a complex and excellently drafted publication. The reviewer thanks very much for the opportunity to have had the pleasure of reading this article and wish you every success in continuing the research and further development!

Reviewer 3 Report

The Manuscript had been improved as per suggestions provided and now the ideas is well represented. However, another round of language editing is required to improve the language.